# Dynamic COVID risk assessment accounting for community virus exposure from a spatial-temporal transmission model

**Yuan Chen**
Department of Biostatistics
Memorial Sloan Kettering Cancer Center
New York, NY 10017
cheny19@mskcc.org

**Wenbo Fei**
Department of Biostatistics
Columbia University
New York, NY 10032
wf2270@cumc.columbia.edu

**Qinxia Wang**
Department of Biostatistics
Columbia University
New York, NY 10032
qw2223@cumc.columbia.edu

**Donglin Zeng**
Department of Biostatistics
University of North Carolina at Chapel Hill
Chapel Hill, NC 27516
dzeng@email.unc.edu

**Yuanjia Wang**
Department of Biostatistics
Columbia University
New York, NY 10032
yw2016@cumc.columbia.edu

## Abstract

COVID-19 pandemic has caused unprecedented negative impacts on our society, including further exposing inequity and disparity in public health. To study the impact of socioeconomic factors on COVID transmission, we first propose a spatial-temporal model to examine the socioeconomic heterogeneity and spatial correlation of COVID-19 transmission at the community level. Second, to assess the individual risk of severe COVID-19 outcomes after a positive diagnosis, we propose a dynamic, varying-coefficient model that integrates individual-level risk factors from electronic health records (EHRs) with community-level risk factors. The underlying neighborhood prevalence of infections (both symptomatic and pre-symptomatic) predicted from the previous spatial-temporal model is included in the individual risk assessment so as to better capture the background risk of virus exposure for each individual. We design a weighting scheme to mitigate multiple selection biases inherited in EHRs of COVID patients. We analyze COVID transmission data in New York City (NYC, the epicenter of the first surge in the United States) and EHRs from NYC hospitals, where time-varying effects of community risk factors and significant interactions between individual- and community-level risk factors are detected. By examining the socioeconomic disparity of infection risks and interaction among the risk factors, our methods can assist public health decision-making and facilitate better clinical management of COVID patients.

## 1 Introduction

The coronavirus disease 2019 (COVID-19) has created several surges of pandemic globally since early 2020 and continues to be a major public health threat. COVID-19 related hospitalizations and

35th Conference on Neural Information Processing Systems (NeurIPS 2021).

deaths have caused immense burdens to the health systems [20, 21]. Therefore, it is crucial to study the community-level risk factors affecting the transmission of the disease and the individual-level risk factors of severe COVID-19 outcomes. We propose a spatial-temporal model for COVID-19 transmissions and a weighted semiparametric time-varying coefficient model for hospitalizations that can integrate multiple levels of information and perform causal inference.

Various models have been proposed to forecast the trend of COVID-19 transmissions [32, 4, 18, 27, 14] based on susceptible-exposed-infectious-recovered (SEIR) model, Gaussian process, or agent-based modeling. Their main goals are on the prediction or forecast accuracy. On the other hand, socioeconomic disparities and spatial variations have been observed in COVID-19 transmissions and individual outcomes [12, 34]. Spatial correlations have been detected for infectious disease transmissions where the disease incidence tends to occur in spatial clusters especially during the initial stages of an outbreak [25, 5]. However, some of these previous forecast models analyze the individual-area temporal trends separately, and do not account for socioeconomic disparity effects.

Our first goal is to propose a joint model for the temporal and spatial patterns across all areas, which allows explaining what accounts for the spatial variability and the spread of infectious disease, and would be more efficient than modeling two trends and for different areas separately. Specifically, the temporal dependence is modeled by taking a convolution of past infection numbers during a transmissible time interval following the framework in [27]. The latent pre-symptomatic disease transmission phase is captured. To allow the spatial dependence, we construct a spatial conditional autoregressive (CAR) model under the Gaussian process to account for the correlation in the infection rates among neighborhood areas and improve the estimation for areas with fewer cases. Area-specific socioeconomic factors are included to account for the heterogeneity in infection rates across regions. For efficient estimation, the proposed temporal and spatial dependence structures are united under a single objective function linked by the infection rates. By examining the spatial disparities and community-level risk factors, our model can facilitate policy decisions and better allocation of healthcare resources to control the disease transmission.

To assess an individual's risk of severe COVID outcomes after diagnosis, risk factors such as demographics and pre-existing medical conditions haven been identified to partially account for the heterogeneity in patients' risks [22, 29, 16, 33]. These existing risk assessment models are solely based on individual-level data. However, community-level risk factors such as neighborhood poverty have been shown to be associated with hospitalization rate [26]. Another important community risk factor is the background risk of virus exposure in the neighborhood measured as the total number of diagnosed and pre-symptomatic infected individuals. This latent virus exposure can be estimated from the proposed spatial-temporal disease transmission model. To fully capture individual risks, we need to consider all these community-level factors.

Furthermore, there are multiple challenges to construct a valid risk assessment model. Firstly, sample selection bias from multiple sources can be present. For example, it is shown that several neighborhoods in New York City (NYC) with high case rates also had high testing rates [26]. As a result, residents from areas with high case rate are more likely to be tested positive and thus more likely to be included in the study sample consisting of only diagnosed patients. A serious challenge is that such selection bias may induce spurious association between risk factor and COVID outcomes and thus misleads interpretation and decision [15]. Secondly, considering the dynamic nature of the pandemic, the effect of the risk factors may not be the same over time. For instance, the number of infected subjects varies substantially throughout the outbreak [26] and the drivers of the variation (e.g., racial disparity) may be different early on in the pandemic versus a later period. Hospital capacity is another important time-varying factor that influences a patient's chance of being admitted, especially during the initial outbreak [11]. However, most existing studies do not consider the selection bias inherited in their study samples [15] and ignore the time-varying effects of the risk factors due to the evolving dynamics of pandemic. Lastly, most of existing risk assessment models lack rigorous calibration and validation procedures [33].

Our second goal is to assess individual risk of severe COVID-19 outcomes accounting for community virus exposure burden and selection bias by a weighted spatial-temporal model. Specifically, we integrate individual-level risk factor data and neighborhood environmental risk factors including the latent time-varying neighborhood virus exposure to better account for patients' heterogeneous risks. Two sets of subject-specific weights are constructed to mitigate the selection biases so that after weighting the study sample could reflect the target population of interest and are not overly selected

from neighborhoods with high infection rates. Furthermore, we build semiparametric models with time-varying coefficients to allow time-dependent effects of hospital capacity and community factors (e.g., number of infectious COVID-19 patients predicted from the spatial-temporal transmission model). The method is applied to model COVID-19 transmission in NYC and assess individual risks using electronic health records (EHRs) of PCR confirmed COVID-19 patients diagnosed at New York Presbyterian Hospital (NYPH). Validation and calibration are conducted to evaluate the proposed method. Our model assists medical decision making and clinical management by integrating risk factors at multiple levels to test which factors significantly influence individual's severe health outcomes after COVID-19 infection.

## 2 Method

### 2.1 Community-level transmission model accounting for spatial and temporal correlation

**Model structure**  We propose a generative model that accounts for both temporal and spatial correlation of disease transmission while allowing heterogeneous infection rates across different areas accounted for by area-specific characteristics (e.g., distribution of minority population and social distancing measures in an area). We follow the framework of the survival convolution model proposed in [27] to account for the disease transmission during a pre-symptomatic phase. We let $a_i(t)$ denote the effective infection rate for the $i$th area ($i = 1, \cdots, n$) on day $t$ defined as

$$a_i(t) = \frac{N_i(t)}{M_i(t)}, \tag{1}$$

where $M_i(t)$ is the number of infected subjects who remain in the transmission chain and can transmit virus to others (including those who are pre-symptomatic or asymptomatic) for the $i$th area on day $t$, and $N_i(t)$ is the number of newly infected subjects for the $i$th area on day $t$. Note that $M_i(t)$ and $N_i(t)$ are both latent processes, and they are more accurate measures of the underlying virus exposure than the daily reported number of diagnosed cases. Let $S(k)$ denote the proportion of individuals remaining infectious and who can transmit disease after $k$ days of being infected, then $M_i(t)$ and $N_i(t)$ are related as

$$M_i(t) = \sum_{k=1}^{C_i} N_i(t-k)S(k+1), \tag{2}$$

where $C_i = \min(t - t_{i0}, \widetilde{C})$, $t_{i0}$ is the unknown day when the first subject is infected in area $i$, and $\widetilde{C}$ is the maximum incubation period (i.e., 14-21 days for COVID-19 [19]). We let $t_{i0}$ be $\widetilde{C}$ days prior to the first reported diagnosis of COVID-19 case. We assume that subjects under quarantine will not be in the transmission chain, and thus $a_i(t)$ reflects the effective transmission rate either due to quarantine or out of infectious period of SARS-COV-2 virus. It follows that the expected number of diagnosed subjects out of transmission chain in area $i$ on day $t$, denoted as $Y_i(t)$, can be calculated based on the latent processes as

$$Y_i(t) = \sum_{k=1}^{C_i} N_i(t-k)[S(k) - S(k+1)]. \tag{3}$$

Note the number of diagnosed patients is the only quantity that is observe. To obtain other quantities, given $t_{i0}$ and $N_i(t_{i0}) = 1$, the expected number of newly infected cases $N_i(t)$ and diagnosed cases $Y_i(t)$ can be updated sequentially based on (2), (1), and (3) if the infection rates $a_i(t)$ and the survival function $S(\cdot)$ are known. Here we model $S(\cdot)$ as the normalized survival function of the exponential distribution, i.e., $S(k) = (e^{-k/\delta} - e^{-\widetilde{C}/\delta})/(1 - e^{-\widetilde{C}/\delta})$, and we set its mean $\delta = 5.2$ following [19].

More importantly, we will model the time-varying infection rate $a_i(t)$ borrowing strength from the neighborhood areas and area-specific time-invariant and time-varying characteristics, e.g., demographics, social vulnerability index, and mobility. Let $\lambda_i(t)$ denote the relative rate of infection in region $i$ compared to a "baseline" expected rate at day $t - 1$ averaged across regions denoted by $\bar{a}(t-1)$, i.e., $\lambda_i(t) = a_i(t)/\bar{a}(t-1)$, where $\bar{a}(t-1) = \frac{1}{n}\sum_{i=1}^{n} a_i(t-1)$, and $\lambda_i(0) = 1$. We denote the baseline average expected number of infections as $E_i(t) = M_i(t)\bar{a}(t-1)$, and assume the actual number of new infections has a mean of $M_i(t)a_i(t)$, or equivalently a mean of $E_i(t)\lambda_i(t)$.

We model the logarithm of the relative infection rate, $Z_i(t) = \log(\lambda_i(t))$, by a spatial conditional autoregressive (CAR) model to account for correlation between regions. Let $\mathbf{Z}(t) =$

$(Z_1(t), \cdots, Z_n(t))^T$ for time points $t = 0, \cdots, T$. Similar to the small area estimation problem [10] where the disease rates for areas with smaller at-risk populations are estimated less accurately, here when estimating $a_i(t)$ in (1), areas with smaller infectious population $M_i(t)$ are subject to more estimation variability. Therefore, to borrow strength from areas with more infectious subjects, we assume $\mathbf{Z}(t)$ follows a Gaussian process that exhibits spatial dependence as

$$\mathbf{Z}(t) \sim MVN(\mathbf{X}_t^T \boldsymbol{\beta}_t, \boldsymbol{\Sigma}_t), \tag{4}$$

where $\mathbf{X}_t$ is a matrix of covariates (e.g., demographics, social vulnerability index, mobility, and include a column of constants of one). The covariance matrix $\boldsymbol{\Sigma}_t$ should reflect heterogeneous variances of estimated infection rates at different locations due to differential baseline number of infectious subjects $M_i(t)$. In addition, the infection rates estimated from smaller populations are more variable and should use more pooling than areas with larger populations. To accommodate these factors, following the spatial rate model used for disease mapping [9, 10] and a spatial CAR model [10], under the condition that $\mathbf{I} - \rho_t \mathbf{H}$ is positive definite, we specify

$$\boldsymbol{\Sigma}_t = \tau_t^2 (\mathbf{I} - \rho_t \mathbf{H})^{-1} \boldsymbol{\Delta}, \text{ where } \boldsymbol{\Delta} = \text{diag}\left(\frac{1}{E_1(t)}, \cdots, \frac{1}{E_n(t)}\right). \tag{5}$$

Furthermore, $\mathbf{H} = (h_{ij})$ has zeros as the diagonal terms, and the off-diagonal terms are specified as $h_{ij} = [E_j(t)/E_i(t)]^{1/2}$ if $j \in G(i)$, and 0 elsewhere. where $G(i)$ indicates the set of areas that share borders with the $i$th area (i.e., the neighborhood of area $i$). In other words, we only borrow information from the neighborhood areas to improve the estimation, without pooling over irrelevant areas. With the covariance matrix specified in (5), we account for the larger variability of small areas with lower expected infection numbers $E_i(t)$ (or equivalent $M_i(t)$). Under this parametrization, it can be shown that $\rho_t$ represents the spatial partial correlation at time $t$ between neighborhood counties $(i, j)$ given other regions, i.e., $\text{corr}(Z_i(t), Z_j(t)|Z_k(t), k \neq i, j) = \rho_t$. This correlation is invariant to the neighborhood structure of counties (i.e., does not depend on $\mathbf{H}$), which is desirable. We provide more explanation of the model structure in the Supplementary material. We will estimate a separate model at each time point $t$. We present an illustrate diagram for the model architecture in Supplementary Figure A.1.

**Estimation** For estimation, we will combine the loss function for the reported daily new cases $R_i(t)$ to the expected number based on (3) and the likelihood of the log relative infection rates $\lambda_i(t)$ as

$$\sum_{i=1}^{n} \sum_{t=t_{i0}}^{T} \left\{\sqrt{R_i(t)} - \sqrt{Y_i(t)}\right\}^2 - \sum_{t=\widetilde{t}}^{T} \log l(\mathbf{Z}(t); \boldsymbol{\xi}_t) + \lambda \sum_{t=\widetilde{t}+1}^{T} ||\boldsymbol{\xi}_t - \boldsymbol{\xi}_{t-1}||_1, \tag{6}$$

where $l(\mathbf{Z}_t; \boldsymbol{\xi}_t)$ is the likelihood under the multivariate Gaussian distribution in (4) with $\boldsymbol{\xi}_t$ denoting the parameters of $(\boldsymbol{\beta}_t, \rho_t, \tau_t^2)$, and $||.||_1$ is the $L^1$ norm. Note in (6), $\widetilde{t}$ denotes the first day of diagnosed case across all areas. In other words, we align the areas according to the relative stage of disease transmission in modeling the spatial dependence. Here the first term in (6) is to evaluate the prediction performance from the latent disease transmission process where the square root transformation is a variance stabilization transformation for count data; the second term is to ensure the spatial smoothness between the areas and account for the heterogeneous infection rates by area-specific factors such as socioeconomics and mobility; and the last term is the fused lasso penalty to ensure the smoothness of the estimated parameters over time. Note that the fused lasso penalty will consequently encourage smoothness in the daily infection rates over time, which reduces variability and is consistent with observed data.

Furthermore, to avoid the high computational burden involved in maximizing the likelihood of the high-dimensional multivariate Gaussian process $\mathbf{Z}_t$, we consider an alternative approach based on optimizing conditional pseudo-likelihood often used to learn Gaussian graphical models. For given $\mathbf{X}_t$ and all $Z_{jt}$'s with $j \neq i$, [6] and [24] showed that the multivariate Gaussian model (4) with covariance matrix of the form (5) implies the conditional normal distributions with

$$Z_i(t)|Z_j(t), j \in G(i) \sim N(\theta_{it}, \tau_{it}^2), \ \theta_{it} = \mu_{it} + \sum_{j \in G(i)} \rho_t h_{ij}(Z_j(t) - \mu_{jt})), \tag{7}$$

where $\mu_{it} = E(Z_i(t)) = \mathbf{X}_{it}^T \boldsymbol{\beta}_t$ and $\tau_{it}^2 = \tau_t^2 / E_i(t)$. Then the objective function (6) becomes

$$\sum_{i=1}^{n} \sum_{t=t_{i0}}^{T} \left\{\sqrt{R_i(t)} - \sqrt{Y_i(t)}\right\}^2 + \sum_{i=1}^{n} \sum_{t=\widetilde{t}}^{T} \log \tau_{it} + (2\tau_{it}^2)^{-1} [Z_i(t) - \theta_{it}]^2 + \lambda \sum_{t=\widetilde{t}+1}^{T} ||\boldsymbol{\xi}_t - \boldsymbol{\xi}_{t-1}||_1,$$

so that the joint likelihood reduces to products of area-wise conditional likelihood and optimization is much easier. To obtain solution, we treat $Z_i(t)$ and $\boldsymbol{\xi}_{t-1}$ as parameters and optimize the objective function by gradient descent implemented by PyTorch [23] on local and cluster CPUs.

Under this model, we can estimate the time-varying between-area correlations of the infection rate and the effects from time-dependent community area-level covariates on the infection rates. With available parameters, the log-relative infection rates $\theta_{it}$ can be estimated from (7) and forecast can be provided based on assumptions of the future epidemic trend (e.g., assuming similar trend over next $d$ weeks as previous $d$ weeks).

## 2.2 Individual risk assessment accounting for selection bias, time-varying effects and community risks

**Weights for controlling selection biases**   Presence of selection bias inherited in the study sample threatens validity of risk prediction models. Many existing models for predicting hospitalization/death once a subject is diagnosed of COVID-19 utilized study samples that only consisted of diagnosed patients. Also the study samples were usually collected from several selected hospitals. In this circumstance, one source of selection bias is that individuals who live closer to a specific hospital are more likely to be included in the study sample. Another source of selection bias is due to the heterogeneous COVID-19 case rates across different neighborhoods, so that individuals from areas with high case rate may be more likely to get tested and diagnosed [26] and thus over-represented in a study sample.

More importantly, since we are interested in in-vestigating whether the prevalence of neighbor-hood virus exposure (i.e., the underlying number of infectious subjects $M_i(t)$) as a measure of community COVID-19 disease burden has an effect on individual's COVID-19 outcomes, the second source of selection bias may induce spurious association. This phenomenon is also known as collider bias [15, 8], which we illus-

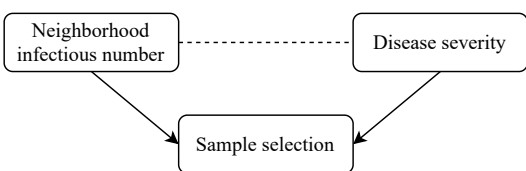

Figure 1: Illustration of the collider bias through a directed acyclic graph. Directed arrows indicate causal effects and dotted lines indicate induced associations.

trate in Figure 1. Note that in this case the risk factor of interest (i.e., measure of community risk) and the outcome of interest (individual COVID outcome severity) are both associated with sample selection. This is because when the study population only consists of diagnosed patients, individuals from neighborhood with high infectious rate are more likely to be diagnosed and thus included in the sample. On the other hand, individuals with more severe symptoms are also more likely to seek testing and get diagnosed, thus prone to be included in the sample. This is particularly the case during the early COVID-19 outbreak when the healthcare facilities were limited and testing was restricted to patients with severe symptoms [2]. Taking these two associations into consideration, conditioning on the selected sample can induce association between the neighborhood infection prevalence and the disease severity even if they were marginally not correlated. Similar collider bias is present for other risk factors of interest (e.g., pre-existing medical conditions).

To address these two selection biases, we propose inverse-probability weighting (IPW) adjustment with two sets of subject-specific weights. We denote $w_{1j}$ as the first weight for the $j$th subject $(j = 1, ..., m)$ constructed as $w_{1j} = \sum_{i=1}^{n} p_i/q_i \, I(S_j = i)$ , where $S_j$ denotes the neighborhood area (e.g., postal area) which subject $j$ comes from, $p_i$ denotes the true population density for area $i$ (i.e., number of residents in area $i$ divided by the total population), and $q_i$ denotes the density of subjects coming from area $i$ $(i = 1, ..., n)$ in the selected sample. Therefore, after weighting, our sample resembles the true spatial population composition. The bias due to association between other time-invariant risk factors and sample selection can be mitigated after addressing the sample selection bias with $w_{1j}$.

The second subject-specific weight aims to remove the association between the time-varying community infection rate and sample selection in order to mitigate the collider bias in Figure 1. Let $w_{2j}$ denote the second weight for subject $j$ defined as

$$w_{2j} = \sum_{t=0}^{T} \sum_{i=1}^{n} \frac{\widetilde{p}_i(t)}{\widetilde{q}_i(t)} I(S_j = i, T_j = t), \quad \widetilde{p}_i(t) = M_i(t) / \sum_{i=1}^{n} M_i(t), \quad \widetilde{q}_i(t) = R_i(t) / \sum_{i=1}^{n} R_i(t).$$

Note $T_j$ is the date when patient $j$ is diagnosed of COVID-19. $M_i(t)$ as defined in section 2.1 is the number of infectious subjects in area $i$ at day $t$ (including those who are pre-symptomatic and have not tested positive), and $R_i(t)$ is the reported number of diagnosed subjects in area $i$ on day $t$. In other words, if there is no causal effect between neighborhood infection prevalence and sample selection, the number of reported diagnosed patients in the sample should be proportional to the number of underlying infectious subjects from each neighborhood on each day. The weights $w_{2j}$ further balance the sample selection from each neighborhood on each day as pandemic continues and remove the bias due to unbalanced numbers of individuals seeking testing.

**High-dimensional feature extraction from EHR data**    EHRs have been increasingly used to assist real world medical decision making. EHRs provide abundant information about patients' medical history, diagnoses, medications, treatment plans, medical procedures, and laboratory test results, etc. To better account for patients' heterogeneity in their existing medical conditions, in addition to the variables derived based on diagnoses and prescribed medications recorded in the EHRs (e.g., diabetes, chronic kidney disease, anticoagulant use), we extract features from the high-dimensional EHRs containing CPT procedure codes and medication prescription codes. Specifically, we filter the top 50 most common procedures and 100 most common medications, and for each of them we record the number of occurrences for each subject. Next, a factor model is fitted to account for the shared variability in the 150 procedures and medications. Specifically, for subject $j$, we let $\mathbf{y}_j - \boldsymbol{\mu}_{\mathbf{y}} = \boldsymbol{\Lambda}\mathbf{z}_j + \boldsymbol{\epsilon}_j$, where $\mathbf{y}_j$ is a vector of length $p$ ($p = 150$) representing the number of each procedure or prescribed medication, $\boldsymbol{\mu}_{\mathbf{y}}$ is the population mean of $\mathbf{y}_j (j = 1, ..., m)$, $\mathbf{z}_j$ is the lower-dimensional latent factors of length $K$, $\boldsymbol{\Lambda}$ is a $p \times K$ matrix of factor loadings, and $\boldsymbol{\epsilon}_j$ is the residuals with $E(\boldsymbol{\epsilon}_j) = 0$ and a diagonal covariance matrix. $\mathbf{z}_j$ and $\boldsymbol{\epsilon}_j$ are assumed to be independent.

These latent factors $\mathbf{z}_i$ serve as efficient dimension reduction technique while capturing the most important common variations in a patient's procedure and medication history. Thus, after fitting the factor model, we use the estimated $\widehat{\mathbf{z}}_i$ as extracted features for each subject. These latent factors are shown to be very strong predictors of the COVID-19 hospital admission in later experiments. At the same time, we can interpret each latent factor by examining the loading matrix $\boldsymbol{\Lambda}$. The factor loadings from the fitted model in the later experiment are explored in Figure D.7 in the Supplementary material and explained in section 3.

**Semiparametric model for time-varying effects**    As discussed in section 1, hospital capacity is a time-varying factor that could influence patient's COVID-19 related outcomes [11, 28]. Additionally, the effect of neighborhood factors (e.g., number or prevalence of infectious COVID-19 patients) vary at different stages of the outbreak. To accommodate these effects, we construct time-varying coefficients in building the risk prediction models using splines. Mathematically, the time-varying coefficient $\alpha(t)$ can expressed as $\alpha(t) = \sum_{k=1}^{K} \phi_k(t)a_k$ where $\phi_k(t)$ are the basis functions (e.g., cubic splines), and we allow the effect of the covariates to vary by day. Then the postulated model for the probability of hospitalization can be written as

$$f(\widetilde{\mathbf{X}}_j, T_j) = \sigma\left(\alpha_0(T_j) + \sum_{l=1}^{d} \alpha_l(T_j)\widetilde{X}_{jl} + \sum_{l=d+1}^{e} \gamma_l \widetilde{X}_{jl}\right),$$

where $\sigma(x) = 1/(1 + e^{-x})$, and $\widetilde{\mathbf{X}}_j = (\widetilde{X}_{j1}, ..., \widetilde{X}_{je})$ are the feature variables for the $j$th subject and we assume the first $d$ variables have time-varying effects. A roughness penalty for each time-varying coefficient is included in the objective function to prevent overfitting and to promote smoothness, which leads to the following objective function

$$\sum_{j=1}^{m} L(y_j, f(\widetilde{\mathbf{X}}_j, T_j)) + \lambda \sum_{l=1}^{d} \int_0^T \alpha_l''(t)dt = \sum_{j=1}^{m} L(y_j, f(\widetilde{\mathbf{X}}_j, T_j)) + \lambda \sum_{l=1}^{d} \int_0^T a_{lk}\phi_{lk}''(t)dt,$$

where $L(y_j, f(\widetilde{\mathbf{X}}_j, T_j))$ is the cross-entropy loss for the binary outcome $y_j$ (e.g., hospital admission) and its fitted probability $f(\widetilde{\mathbf{X}}_j, T_j)$ for the $j$th subject. $\alpha_l''(t)$ is the second derivative of the $l$th time-varying coefficient. Generalized cross validation is used to choose the tuning parameter $\lambda$. Asymptotics for penalized splines have been previously established [7, 31]. Therefore, we can construct confidence intervals to evaluate the uncertainty of the model estimates and the predicted individual risks.

## 3 Experiments

### 3.1 Transmission model accounting for spatial dependence

New York City (NYC) was the epicenter of COVID-19 in the United States during spring 2020 [26]. We applied the spatial-temporal disease transmission model described in section 2.1 to the COVID-19 data in NYC from early March to end of July 2020, which correspond to the first wave of the outbreak in NYC. ZIP code-level daily reported new cases, social vulnerability index (SVI, e.g., minority percentage) and mobility data from opted-in users (percentage of individuals shelter-in-place captured by their mobile devices) were used to model disease transmission process. We plot the covariates in Supplementary Figure D.1 where heterogeneity across ZIP areas in NYC are observed. The data sources and preprocessing were described in the Supplementary material.

We identified a significant spatial correlation of COVID-19 transmission in NYC from the spatial-temporal model ($\rho$= 0.109, 95% CI: (0.102, 0.117)). To visualize the COVID transmission in NYC, in Figure 2 we show the estimated infection rate in each area at a few representative time points. The first surge of the pandemic in NYC was characterized by the highest infection rates during mid to late March and a much lower rate during May with a slight uptick from June, corresponding to the NYC stay-at-home order which was in full effect from March 22 to June 8, 2020. Change points of infection rates (e.g., peak around March 19 and rebound around June 11) were observed in Supplementary Figure D.2, which shows the infection rates in each neighborhood from late February to the end of July. Spatial disparity and correlation at the ZIP code level were also observed in the figure. Furthermore, we detected a significant racial disparity where neighborhoods with a denser minority population suffer a higher risk of COVID-19 infection. The effect coefficient $\beta_t$ has a decreasing trend and was estimated to be 0.046 (95% CI: (0.026, 0.066)) on March 10, 0.030 (95% CI: (0.012, 0.048)) on March 19, 0.020 (95% CI: (0.017, 0.023)) on June 11, and 0.017 (95% CI: (0.015, 0.018)) on July 11. The confidence intervals were constructed based on permuting residuals, and we describe the procedure in the Supplementary material. The shelter-in-place percentage was not significantly associated with infection rates. Since this mobility variable was collected from only opted-in individuals, more accurate and representative measure is required in order to improve the estimation. Additionally, we show the observed and estimated daily new COVID-19 cases for each ZIP code area in Supplementary Figure D.5. The fitted curves captured the central trend with a smoother fit. For a few zip-code areas, the number of cases was under-estimated, especially around the peak period due to multiple reasons. First, the data variability was high in some zip areas where there were abnormal/extreme spikes. Those spikes were primarily due to data backlog and sudden "data dumps" instead of true case rises [1]. Since our model encourages a smoother fit by incorporating penalty terms for consecutive days, those abnormal spikes cannot be (and perhaps should not be) captured. Second, since we pool information from neighborhood areas to infer infection rates, when the local spatial dependence is not very strong, pooling information may affect the area-specific case estimation. However, for the majority of zip areas, the proposed spatial model is beneficial. Future studies may extend the current model to allow more flexible spatial dependence structure, e.g., community-specific dependence parameters.

**Experiments on simulated data** We conducted additional numerical studies using simulated data to evaluate the performance of our proposed method in recovering the true time-varying coefficients accounting for the spatial and temporal correlation in COVID-19 transmission. Date were simulated based on observed NYC ZIP level data where minority percentage was included as the area specific feature variable. Area specific time-varying infection rates were simulated based on the Gaussian process model under the specified true parameters, where the true $\beta_0(t)$ ranged from $-0.05$ to $0.05$, $\beta_1(t)$ ranged from 0 to 0.06, $\tau(t)$ ranged from 0.01 to 0.08, and $\rho$ was set to 0.012 to mimic the parameters estimated from the real data. The number of infections and reported cases were then simulated based on the proposed disease transmission model. The simulation code are provided in the Supplementary materials. We considered two scenarios, one for the 44 ZIP-code defined areas in Manhattan only and one for the 176 ZIP areas in all boroughs of NYC.

We replicated the simulation experiments 100 times. The rooted mean squared errors (RMSEs) in estimating the time-varying parameters is presented in Supplementary Figure D.3, where the RMSEs were calculated across all time points and the figures show variability from the experiment replications. For the time-invariant parameter $\rho$, we obtained a mean estimate of 0.107 (RMSE of 0.015 from 100 replications) under 44 areas and a mean estimate of 0.116 (RMSE of 0.010) under

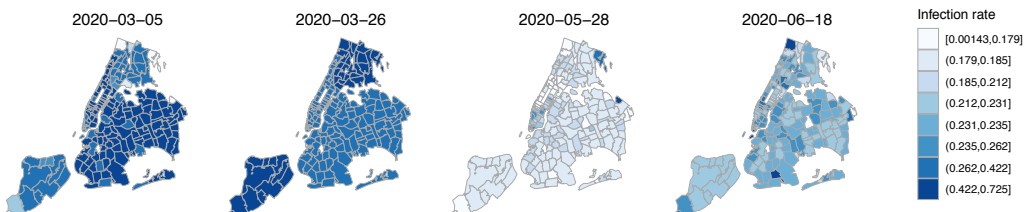

Figure 2: Estimated infection rates for neighborhood areas in NYC at selected dates.

176 areas. Therefore, the parameters of interest (i.e., $\beta_0(t)$, $\beta_1(t)$, and $\rho$) are accurately estimated, and the accuracy increases with the increase of number of areas (lower RMSEs under 176 areas compared to 44 areas). Furthermore, in Supplementary Figure D.4 we present the RMSEs in estimating the reported diagnosed cases based on the estimated parameters from 100 replications. Small RMSEs were obtained compared to the large daily reported case numbers (an average of 31 cases across time and areas with an average maximum of 296 cases across replications in the scenario of 176 areas). These simulation results demonstrate that the proposed method and learning algorithm can recover the true parameters accounting for area heterogeneity and spatial correlations as well as the underlying transmission process of COVID-19.

## 3.2 Individual risk assessment accounting for selection biases and time-varying effects

We applied the proposed bias-corrected semiparametric risk assessment model introduced in section 2.2 to 6911 subjects who were diagnosed of COVID-19 from two New York-Presbyterian hospitals from March 9 to July 6, 2020. Among these patients, $51.8\%$ were female, the median age was 60, and there were a total of 3676 hospital admissions and 717 deaths. The EHR study protocol (with a study end date of July 6) was approved by the IRB.

In building the risk assessment model, we integrated the individual EHR data with the neighborhood risk factors on two levels, spatially and temporally. Specifically, for each subject, to evaluate the risk due to the background risk of virus exposure at infection, we incorporated the predicted number of infectious subjects (i.e., $M_i(t)$ acquired from the transmission model in sec-

Table 1: Fitted coefficients of hospitalization risk among patients diagnosed with COVID-19.

|  | Weighted | Unweighted |
| --- | --- | --- |
| Age (standardized) | 0.18 (0.1, 0.25)*** | 0.4 (0.33, 0.46)*** |
| Female | -0.29 (-0.43, -0.15)*** | -0.33 (-0.45, -0.21)*** |
| Asian vs. White | -0.82 (-1.41, -0.23)** | -1.18 (-1.7, -0.67)*** |
| Black vs. White | -0.08 (-0.38, 0.22) | -0.02 (-0.22, 0.18) |
| Minority vs. White | -0.42 (-1.62, 0.78) | -1.23 (-2.1, -0.36)** |
| Hispanic: Yes vs. No | 0.75 (0.49, 1.01)*** | 1.06 (0.89, 1.24)*** |
| Diabetes | 0.47 (0.03, 0.92)* | -0.11 (-0.56, 0.34) |
| Chronic kidney disease | 1.44 (0.2, 2.68)* | 0.76 (-0.39, 1.91) |
| Respiratory disease | 2.46 (1.34, 3.58)*** | 1.69 (0.9, 2.48)*** |
| Cancer | 1.31 (-0.03, 2.64) | 0.25 (-0.69, 1.18) |
| Mental illness | 0.86 (-0.52, 2.23) | 0.87 (-0.17, 1.91) |
| Anticoagulant use | 0.35 (-0.12, 0.81) | 0.08 (-0.37, 0.54) |
| Procedure & medication F1 | 0.1 (0.03, 0.18)** | -0.04 (-0.11, 0.03) |
| Procedure & medication F2 | 1.3 (1.09, 1.5)*** | 1.11 (0.92, 1.29)*** |
| White × Minority (%) | 0.85 (-0.73, 2.44) | 2.75 (1.75, 3.74)*** |
| Asian × Minority (%) | 0.2 (-1.91, 2.31) | -0.86 (-3.13, 1.41) |
| Black × Minority (%) | 1.85 (0.71, 2.99)** | -0.91 (-1.91, 0.1) |
| Minority × Minority (%) | -4.44 (-9.61, 0.73) | -3.03 (-7.73, 1.68) |
| Hispanic Yes × Minority (%) | 1.77 (0.86, 2.68)*** | 0.21 (-0.67, 1.1) |
| White × Multi-unit (%) | 1.19 (-0.71, 3.08) | -0.95 (-2.56, 0.66) |
| Asian × Multi-unit (%) | -1.07 (-3.82, 1.67) | 0.5 (-2.36, 3.36) |
| Black × Multi-unit (%) | 0.31 (-1.1, 1.73) | 5.64 (4.12, 7.16)*** |
| Minority × Multi-unit (%) | -0.59 (-7.16, 5.97) | 0.05 (-8.11, 8.21) |
| Hispanic Yes × Multi-unit (%) | 2.46 (1.02, 3.9)*** | 5.05 (3.58, 6.52)*** |

\* for p-value < 0.05, ** for p-value < 0.01, *** for p-value < 0.001.

tion 2.1) in his/her neighborhood 7 days prior to diagnosis (on average 5 days of incubation [19] and we consider 2 days of time lapse before diagnosis) as a covariate in the risk assessment model. This measure is the total prevalence of COVID-19 infections including both symptomatic and pre-symptomatic subjects who have not yet tested positive, and thus more accurately assess community COVID-19 risk. Additionally, we matched each individual with the time-invariant neighborhood information (e.g., the neighborhood social vulnerability index) based on their billing ZIP codes. Interactions between individual-level race and ethnicity and neighborhood social vulnerability were examined. Time-varying intercept and time-varying coefficients for neighborhood infection prevalence, minority percentage, multi-unit living percentage were constructed.

To visualize the selection biases in the study sample, we plot in Supplementary Figure D.6 the sample frequency and heterogeneous infection patterns across ZIP code areas in NYC. To assess the utility of the weights derived in section 2.2 accounting for the selection biases, we fit both weighted and unweighted models to assess hospitalization risk among diagnosed COVID-19 patients.

For the weighted model, we multiplied the two proposed weights, performed standardization, and then truncated at 5th and 95th percentiles to eliminate extreme values. Model discrimination and

calibration were evaluated on independent subjects through 4-fold cross-validation. Experiments were conducted using R package "mgcv" 1.8.34 (license: GPL-2 | GPL-3) [30] on local CPUs.

Results in Table 1 suggests that accounting for the selection biases led to more meaningful model results than the unweighted model. Age, sex, race, medical conditions such as diabetes, chronic kidney disease, respiratory disease are shown to be strong predictors for hospitalization in the weighted model, which is consistent with the literature and CDC guidance [16, 3]. If we ignore the selection biases, some of these coefficients were not statistically significant and some were in the unexpected direction. For example, having diabetes was found to be associated with lower risk of hospitalization in the unweighted model. Additionally, we detected interactions between individual-level characteristics and neighborhood-level factors. Among African Americans and Hispanics, higher neighborhood minority percentage is associated with a higher risk of COVID-19 hospitalization. Hispanics from higher level multi-unit living environment in NYC were more likely to be hospitalized after being diagnosed of COVID-19.

The two factors we learned from the high-dimensional CPT codes and medication codes in the EHRs were shown to be strongly associated with severe COVID-19 outcomes. From the factor loadings in Figure D.7 in the Supplementary material, the first factor loads on almost all medications and procedures and represents the general medical burden. The second factor is more sparse and more predictive of hospitalization. It is mainly characterized by pain relievers, ancillary drugs for surgery, drugs for controlling side effects of cancer treatments, and glucose for hypoglycemia. Hence, the result indicates that history of surgery, cancer, and hypoglycemia are strongly related to an individual's higher risk of adverse COVID-19 outcomes, which is consistent with the literature [13, 16].

We show the fitted time-varying coefficients in Figure 3. After accounting for individual risk factors, diagnosed patients had a higher probability of hospital admission at later time periods in the first surge of COVID-19 outbreak in NYC (Figure 3b). This may be explained by the limited hospital capacity during the initial outbreak [17], and more health care resources were available towards May 2020. However, the death rate decreased consistently, indicating mortality is higher when hospital resource is scarce [28] (Figure 3d). On the other hand, higher neighborhood virus exposure (indicated by number of infectious subjects) is associated with higher risk for both hospitalization and mortality especially during the post-peak period of the outbreak (Figure 3a,3c). It suggests controlling neighborhood community infection prevalence may have an effect on preventing an individual's severe COVID-19 outcomes above and beyond an individual's own risk factors.

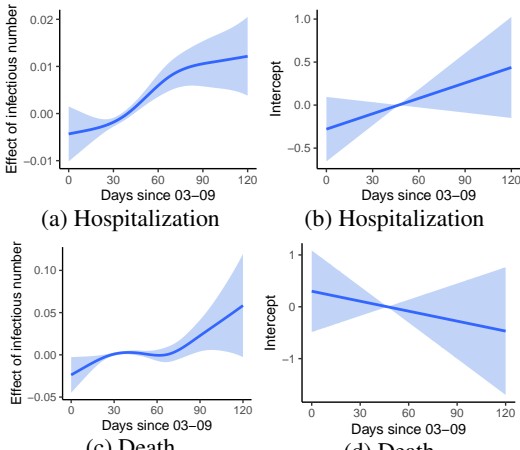

(a) Hospitalization     (b) Hospitalization

(c) Death     (d) Death

Figure 3: Time-varying coefficient of neighborhood infection prevalence (including pre-symptomatic infections) and intercept (with 95% confidence bands) on the risk of hospitalization and death respectively for subjects diagnosed with COVID

To examine the marginal effect of the background virus exposure at infection, we further show in Figure 5 the predicted time-varying risk for hospitalization under different prevalence of neighborhood infections. When the neighborhood COVID-19 prevalence is low, we observe a decreasing risk of hospitalization which gradually tends stable after the peak of outbreak. However, the risk for hospitalization is consistently increasing if the neighborhood infection prevalence is persistently high.

We calibrated and validated the fitted model through a 4-fold cross validation, and results were averaged from all validation sets and presented in Figure 4. We evaluated the two fitted models on the validation sets after correcting selection biases via the two sets of weights $w_{1j}$ and $w_{2j}$. A higher area under the receiver operating characteristic (ROC) curve is observed for the proposed method that addressed the selection biases. We assessed model calibration by comparing weighted mean predicted risk with mean observed risks grouped by deciles of the predicted risk on the validation sets. Much higher calibration is observed for the weighted model where the fitted regression line is very close to the diagonal line while the unweighted model fails to calibrate on the validation sets. These results demonstrate the generalizability of the proposed weighting strategy.

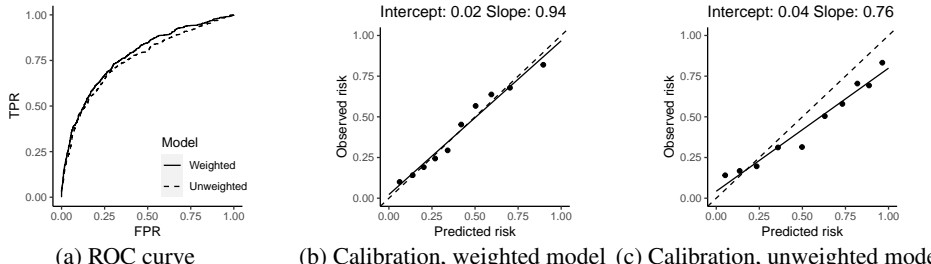

(a) ROC curve     (b) Calibration, weighted model   (c) Calibration, unweighted model

Figure 4: ROC and calibrations of the weighted and unweighted individual risk model on the validations sets. In (b) and (c), a regression line (the solid line) is plotted for the observed vs. predicted risks (the closer to the diagonal dash line is more desirable), and the fitted intercept and slope are reported in the figure.

## 4 Discussion

In this work, we propose a spatial-temporal model for COVID-19 transmission and a bias-corrected semiparametric time-varying coefficient model for assessing individual risk for severe COVID-19 outcomes to integrate multiple levels of information and draw causal inference. The spatial and temporal dependence for disease transmission and heterogeneous time-varying infection rates across areas are accommodated. The relative strength of community-level and individual-level risk factors on explaining severe COVID outcomes are assessed accounting for evolving dynamics of the pandemic.

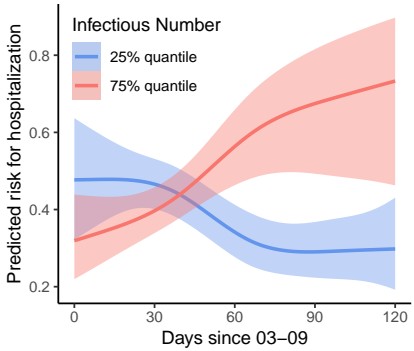

Figure 5: Estimated risk of hospitalization over time under different prevalence of neighborhood infections (including pre-symptomatic infections as oppose to only reported infections). Other covariates fixed at the mean age of 59, male, white, non-hispanic, no diabetes or other comorbidities, median value of the factor scores for procedure and medication, median value of the neighborhood minority level and multi-unit living level.

A limitation of this work is that the individual-level EHR data were collected from two hospitals in NYC. Larger and more general populations can be considered to further test our method. COVID-19 vaccines were not available during the time period of our EHR data. Future studies can consider including vaccination information in transmission and disease risk modeling.

Here we discuss the potential negative societal impacts. In this work, we account for the potential biases in the sample selection using carefully designed weights so that after weighting our sample could represent the general population of interest and avoid spurious association. Even though we have made best attempts to design the weights to balance the sample selections at the ZIP code level, there could be residual imbalances in specific socioeconomic factors because the weights were not explicitly defined by them. Additionally, although we have incorporated many risk factors to account for the individual health outcomes, individual-level behavioural measures such as masking were not included due to data availability. Therefore, when interpreting the results, potential unmeasured confounding shall be noted.

We detected a significant spatial correlation of COVID-19 transmission in NYC and a significant racial disparity where neighborhoods with a denser minority population suffer higher risk of COVID-19 infection. We also identified a significant interaction between individual's race and community-level SVIs for COVID-19 hospitalization beyond an individual's other risk factors (e.g., age, co-morbidities). These findings suggest an intricate interplay between individual-level and community-level risks for COVID-19, and community-level risk factors are non-ignorable even after accounting for the individual-level characteristics. The significant interactions and time-varying effects can facilitate precision public health decision-making at both community- and individual-level, i.e., to inform when, which population, and in what communities should we target the intervention to better reduce the hospitalization burden. For example, our results suggest that it can be beneficial to target the intervention to Hispanic and black communities living in areas with dense minority populations and target the multi-unit living buildings specifically for the Hispanic population. To prepare for future pandemics, a comprehensive approach targeting both individual health and community risks is highly desirable.

## Acknowledgments and Disclosure of Funding

This research is supported by U.S. NIH grants CA008748, NS073671, GM124104, and MH123487.

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
