# Supplementary Material for

# Dynamic COVID risk assessment accounting for community virus exposure from a spatial-temporal transmission model

**Yuan Chen**
Department of Biostatistics
Memorial Sloan Kettering Cancer Center
New York, NY 10017
cheny19@mskcc.org

**Wenbo Fei**
Department of Biostatistics
Columbia University
New York, NY 10032
wf2270@cumc.columbia.edu

**Qinxia Wang**
Department of Biostatistics
Columbia University
New York, NY 10032
qw2223@cumc.columbia.edu

**Donglin Zeng**
Department of Biostatistics
University of North Carolina at Chapel Hill
Chapel Hill, NC 27516
dzeng@email.unc.edu

**Yuanjia Wang**
Department of Biostatistics
Columbia University
New York, NY 10032
yw2016@cumc.columbia.edu

In this supplementary material, we present an illustrative diagram of the spatial-temporal disease transmission model, details about the data sources and data preprocessing, and additional numerical results for the Experiment session in the manuscript.

## A   Diagram of the spatial-temporal disease transmission model

We illustrate in the following diagram the dependence of the hidden process $M(t)$, $N(t)$, $a(t)$, and the observed process $Y(t)$ and $X(t)$ in the proposed spatial-temporal disease transmission model for COVID-19.

35th Conference on Neural Information Processing Systems (NeurIPS 2021), Sydney, Australia.

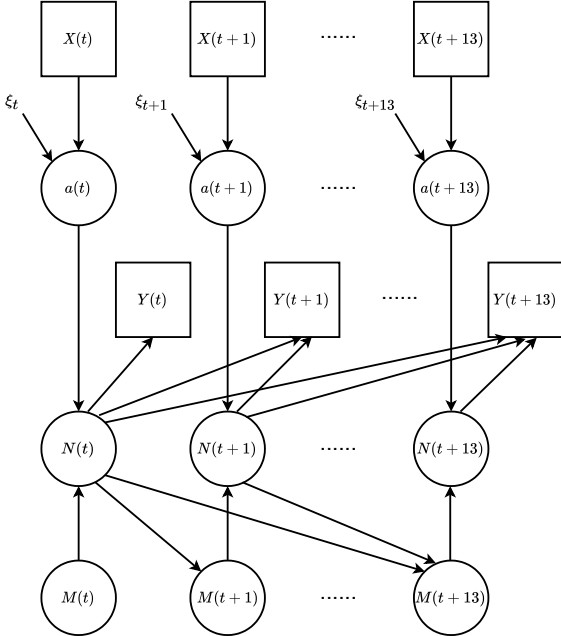

Figure A.1: Illustration diagram of the spatial-temporal model for one consecutive 14 days (the maximum incubation period of COVID-19). $M(t)$: number of infected subjects who remain in the transmission chain and can transmit virus to others (including those who are pre-symptomatic or asymptomatic) on day $t$. $N(t)$: number of newly infected subjects on day $t$. $Y(t)$: number of diagnosed subjects out of transmission chain on day $t$. $a(t)$: infection rate on day $t$, which depends on area characteristics $X(t)$ and spatial-temporal transmission model parameters $\xi_t$.

## B  More details for the spatial-temporal model

Recall that we specify the correlation matrix in the spatial-temporal model to be the following structure

$$\boldsymbol{\Sigma}_t = \tau_t^2 (\mathbf{I} - \rho_t \mathbf{H})^{-1} \boldsymbol{\Delta}, \text{ where } \boldsymbol{\Delta} = \text{diag}\left(\frac{1}{E_1(t)}, \cdots, \frac{1}{E_n(t)}\right).$$

The choice of $\Sigma_t$ was motivated by the spatial rate model in the disease mapping literature (please refer to Chapter 4.2.6 in Statistics for Spatio-Temporal Data by Cressie and Wikle). In the disease mapping literature, such covariance structure is designed to facilitate inferring the disease rate for small areas (areas with a smaller population). We adopted this structure in our case to help infer the infection rate $\alpha_i(t)$ for "small areas" (areas with a small number of infectious subjects $M_i(t)$ and hence a smaller expected new infection number $E_i(t)$). The idea is that for small areas, the observed rates are more variable because the variance for the binary variable infection rate is $p(1-p)/M$. Hence, we wanted to borrow strength from areas with large $M$ (or equivalently large $E$). In the specified covariance matrix, this variance heterogeneity is expressed through $\tau_t^2 \Delta$, where $\Delta$ is a diagonal matrix with $1/E_i(t)$ on the diagonals. The middle part $(\mathbf{I} - \rho_{\mathbf{t}} \mathbf{H})^{-\mathbf{1}}$ is constructed to ensure the partial correlation between two neighborhood areas given the other neighborhood areas is $\rho_t$ so that we can interpret this parameter as the spatial correlation. This can be derived by the property of Gaussian distribution that the inverse of the Gaussian covariance matrix is the partial correlation.

## C  Data sources and data preprocessing

The ZIP code-level daily reported new cases were obtained from the NYC Department of Health and Mental Hygiene (https://www1.nyc.gov/site/doh/covid/covid-19-data.page#epicurve), and county-level daily observed new cases were collected from the website of Johns Hopkins Coronavirus Resource Center (https://coronavirus.jhu.edu/us-map). The NYC neighborhood social vulnerability index (SVI, e.g., minority percentage, multi-unit living percentage) on the FIPS code (census

block group) level were obtained from the Centers for Disease Control and Prevention (CDC, https://www.atsdr.cdc.gov/placeandhealth/svi/index.html). These are all publicly available databases. ZIP code level SVIs were then calculated by taking the weighted average of the FIPS code level data weighted by FIPS population. We collected daily FIPS code level social distancing metrics data from Safegraph (https://www.safegraph.com/). The data were generated using a panel of GPS pings from anonymous opted-in individual mobile devices. We calculated the percentage of shelter-in-place subjects by dividing the number of completely-at-home devices by the total number of devices in each FIPS area. The percentage of shelter-in-place subjects in each ZIP area was calculated by mapping FIPS areas to ZIP areas.

## D    Additional numerical results

**Community-level Transmission model**    In Figure D.1, we plot the covariates used in the disease transmission model to account for area heterogeneity. In Figure D.2, we plot the estimated infection rates across all ZIP code areas in NYC from late February to the end of July, 2020. In Figures D.3 - D.5, we show the additional results for the community-level COVID transmission model in estimating the true parameters of interest and recovering the number of reported diagnosed cases.

Here we describe the procedure to construct the confidence intervals for the parameters in the spatio-temporal model. We subtracted the estimated from the observed number of daily new cases to get the residuals. We permuted residuals of each area across time, more specifically, exchanging residuals within days 1-14, 15-98, 99-160 as the variance of the residuals had a similar scale within each period. We did not permute across areas as it might disturb spatial correlation. We permuted 100 times and fit a separate model to each set of permuted data. Based on the permutation results, we estimated the variance of the parameters and constructed confidence intervals as estimate +/- 1.96*SE.

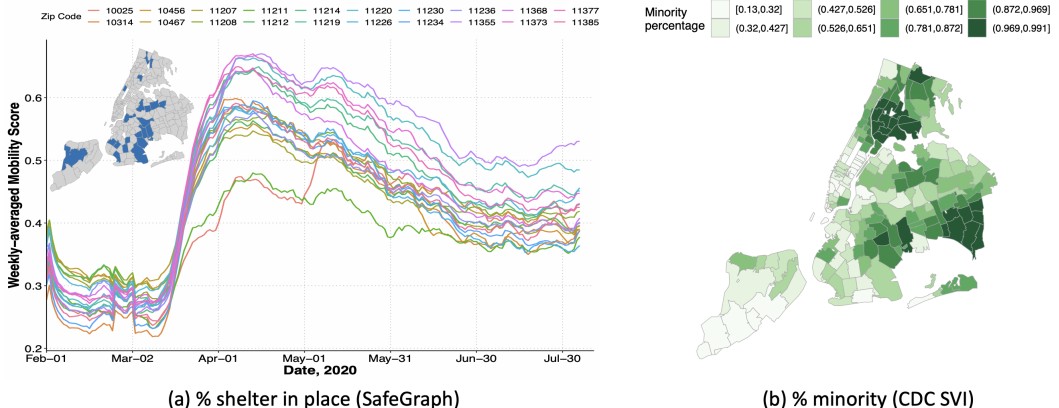

(a) % shelter in place (SafeGraph)        (b) % minority (CDC SVI)

Figure D.1:  (a) SafeGraph mobility measure: % users "shelter-in-place" in 20 most populated zip codes (averaged over past week); (b) CDC % minority in NYC.

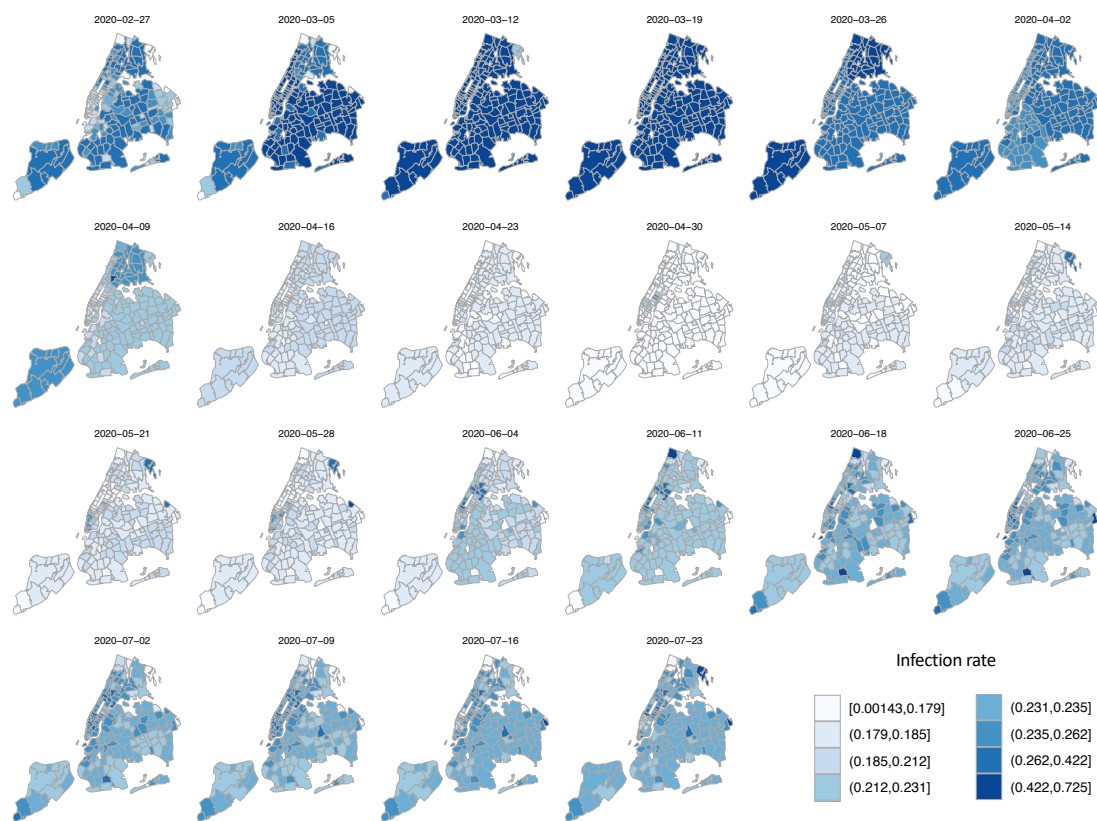

Figure D.2: Time-varying infection rates estimated from the spatial-temporal transmission model for all ZIP code areas in NYC.

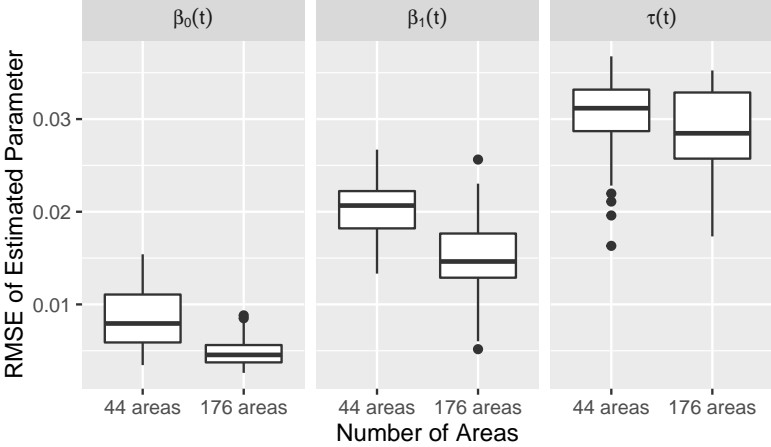

Figure D.3: Rooted mean squared errors (RMSEs) in estimating the time-varying parameters in the spatial-temporal disease transmission model from 100 replicated simulation experiments on 44 or 176 areas. RMSEs were calculated across all time points and the figures show variability from the experiment replications.

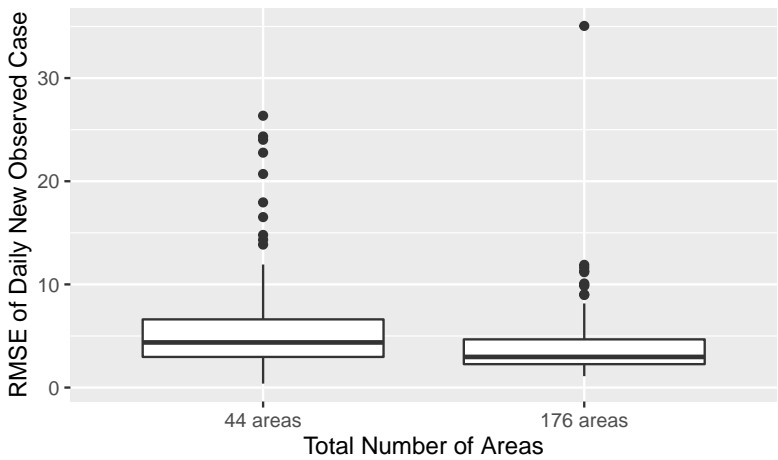

Figure D.4: Rooted mean squared errors (RMSEs) in estimating the diagnosed cases based on the spatial-temporal disease transmission model from 100 replicated simulation experiments on 44 or 176 areas. RMSE value was calculated over all areas and time points in each replication.

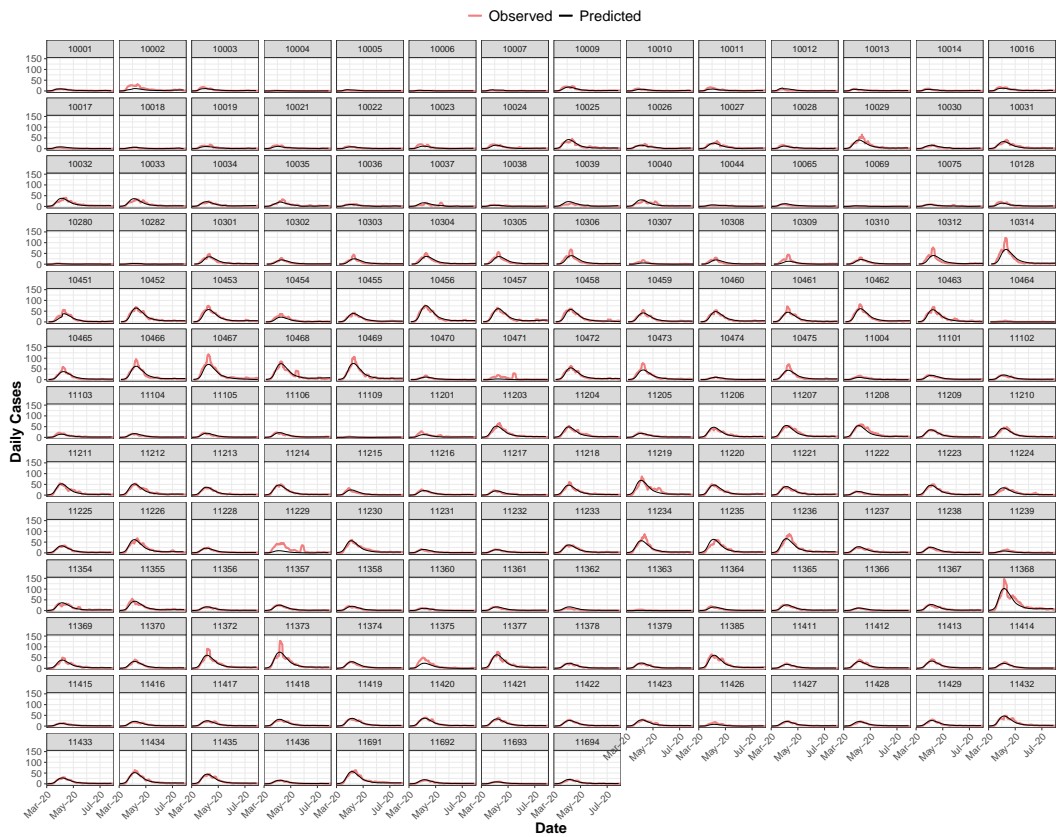

Figure D.5: The predicted vs. observed daily diagnosed COVID-19 cases for all ZIP code areas in New York City based on the spatial-temporal transmission model.

**Individual-level model for risk assessment**  In Figures D.6 and D.7, we present additional results from the individual-level risk assessment models.

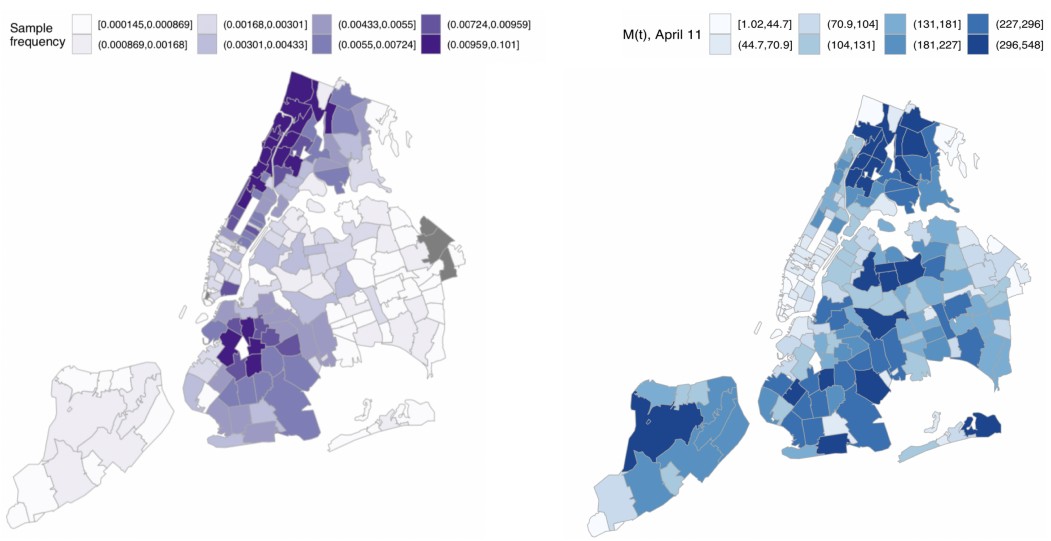

Figure D.6: (a) Sample frequency in the EHR data by ZIP code areas in NYC; (b) Virus load indicated by the estimated underlying infectious subjects in each ZIP code area in NYC

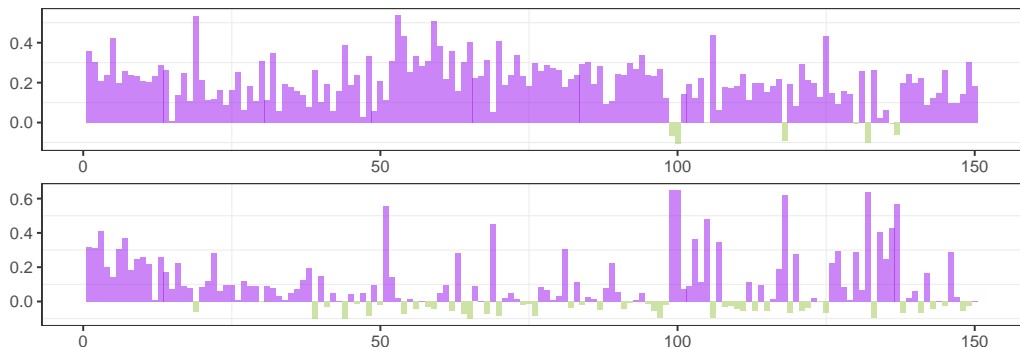

Figure D.7: Loadings of the two factors from the factor analysis of the top 50 procedures (first 50 bars) and 100 medications (last 100 bars) using the NYPH EHRs. The first panel is for factor 1 (general health), and the second penal is for factor 2 (specific conditions).