# OpenReview forum: "Dynamic COVID risk assessment accounting for community virus exposure from a spatial-temporal transmission model"
_NeurIPS.cc/2021/Conference — NeurIPS 2021 Poster_

### Official Review · Reviewer_e3RN · 2021-07-08

**Rating:** 7
**Confidence:** 3

**Summary:**

This work proposes (a) a community-transmission model, and (b) an individual risk-assessment model for COVID-19. The model is general enough to be applicable to any infectious disease. The model for (b) is enriched by inferred variables from (a) by (i) using them as covariates and (ii) addressing the selection bias in the data used for (b).

### community-transmission model

First, this work links the temporal model for forecasting the number of infectious cases with a spatial model to account for disease propagation between regions.
The connection is made via modelling the relative infection rate as a Gaussian Process parameterized by a covariance matrix that depends on the expected number of infectious cases in the neighbouring regions.
The simulation results in the Appendix demonstrate that the model is able to uncover the true underlying parameters.
This spatiotemporal model is then fitted to the reported number of daily COVID-19 cases in NYC from March to July, and the results show a substantial spatial correlation in disease propagation.

### individual risk-assessment model

Finally, an individual risk-assessment model is constructed using the inferred variables from the above fitted spatiotemporal model.
The data used to build this model is from two hospitals in New York.
The authors addressed the issue of selection bias in the data via subject-specific weights by making the sample representative of true underlying prevalence and spatial population.
The results seem to suggest a superior fit to the data compared to the unweighted model.



**Ethics Review Area:**

["I don’t know"]

**Limitations And Societal Impact:**

The paper has discussed the potential pitfalls of their modelling approach.

**Main Review:**

### Writing

The paper could have been written or structured better. It was hard to follow through the description at times, for example, it was difficult to follow the description in Methods at places. Some equations use variables that are explained after quite a while, for example, $\rho_t$ in eq. 5. Some variables are not explained at all, for example, $\tau_t$ in eq.5.

It was difficult to connect the two models. See my last comment in Experiments.

### Novelty

There have been several discussions in the community about growing inequalities due to the COVID-19 pandemic.
As pointed out in the introduction, and to the best of my knowledge, there aren’t studies that forecast infection or assess individual risks by incorporating socio-economic and spatial factors even though these factors are deemed crucial in the transmission of viruses.
This research attempts to mathematically model and assess the correlation between various socioeconomic variables to individual disease severity and spatial propagation.
I think this work could be useful both practically and to make further progress in the field of modelling infectious diseases with socioeconomic factors.

### Methods

1. Line 113: Survival function seems similar to assuming characteristics of an infectious individual’s likelihood to infect susceptible individuals. Following this line of thought, can the survival function follow the same distribution as viral load curve, i.e., a gamma distribution with the same mean (akin to the modelling of infectiousness in most published studies)? Do the authors think that will result in better estimations?
2. Eq. 5: Is the choice $\Sigma_t$ motivated from prior work? Can the authors briefly describe the intuition behind such a choice in Appendix? It is hard to find this in the resources referenced.
3. Line 135: Can the denominator be 0? If so, what should be $h_{ij}$ in that case? Will this not happen because the summation in Eq. 6 is from the day of observing a min of 1 case across all areas?
4. Line 142-143: I don’t get how $\rho_t$ is a scalar given that it depends on i,j,k. Can the authors provide proof of this statement or refer to the relevant text in the referenced papers? Even an intuitive explanation might suffice for the readers.
5. Line 173: Can the authors suggest a way to modify Eq. 7 to account for the reversal of trend (e.g., waves of infections after some decline in cases)?
6. Line 233 - 245: Can the authors comment whether PCA could be as useful for dimensionality reduction as the proposed factor model? Did they try PCA? Are there results on PCA as well?
7. Line 246-263: What is the relation between $\tilde{\mathbf{X}}_j$ and $\hat{\mathbf{z}}_j$?
8. Eq.6: Is the first term any different from $(R_i(t) - Y_i(t))^2$? It doesn't seem like, but if so, can the authors explain so that the reader is aware?

### Experiments

1. Line 282: Can it be shown visually? For example, show a spatial map of the density of the minority population. This will help the reader understand this sentence better.
2. Line 283: Is it $\rho_t$ instead of $beta_t$?
3. Line 287: The procedure to construct confidence bounds is not explained in the supplementary material.
4. It is not entirely clear how exactly the spatio-temporal model in section 2.1 is related to the individual-risk model in section 2.2. The connection is through the correction for sampling bias ($w_{2j}$) and using $M(t-7)$ as a covariate. For a complete understanding of this connection, can the authors compare their regression results with a risk model that includes only $w_{1j}$ and doesn’t use $M(t-7)$ as a covariate?

### Minor comments

1. Line 244: Figure C.4 (not C.3) corresponds to the factor loadings
2. Line 291: Figures C.1 and C.2 are exactly the same. I think you meant Figure C.3.
3. Line 308: IRB could mean different things to people from different countries. Can the authors please provide a link to this organization?

### Broader comments

Can the authors comment on what would be the pitfalls or advantages of jointly training the two models? Does it sound reasonable to do so?

Can the authors comment on whether their model would be appropriate to forecast waves of infection? Will it require additional variables in modelling?




**Time Spent Reviewing:**

10

---

> ### Author Response · Authors · 2021-08-10
> **Author Response**
>
> We thank you for your extremely careful reading, all positive comments, and constructive suggestions. Our pointwise responses are as follows.
>
> For writing, we will better structure the paper especially for the descriptions in the Methods as you suggested to ensure better readability. The explanation for the parameters in the covariance matrix is provided in Methods Response 2.
>
> $\textbf{Methods}$
>
> $\textbf{Response 1}$: Yes, different survival functions can be used to characterize an infectious individual’s likelihood to infect susceptible individuals. We used the exponential function following previous literature (Wang et al. 2020). We thank you for drawing the connection with the viral load curve.
>
> $\textbf{Response 2}$: Yes, the choice of $\Sigma_t$ was motivated by the spatial rate model in the disease mapping literature (please refer to Chapter 4.2.6 in Statistics for Spatio-Temporal Data by Cressie and Wikle). In the disease mapping literature, such covariance structure is designed to facilitate inferring the disease rate for small areas (areas with a smaller population). We adopted this structure in our case to help infer the infection rate $\alpha_i(t)$ for “small areas” (areas with a small number of infectious subjects $M_i(t)$ and hence a smaller expected new infection number $E_i(t)$). The idea is that for small areas, the observed rates are more variable because the variance for the binary variable infection rate is $p (1-p) / M$. Hence, we wanted to borrow strength from areas with large $M$ (or equivalently large $E$). In the specified covariance matrix, this variance heterogeneity is expressed through $\tau_t^2 \Delta$, where $\Delta$ is a diagonal matrix with $1 / {E_i(t)}$ on the diagonals. The middle part $(\bf{I}-\rho_t H)^{-1}$ is constructed to ensure the partial correlation between two neighborhood areas given the other neighborhood areas is $\rho_t$ so that we can interpret this parameter as the spatial correlation. This can be derived by the property of Gaussian distribution that the inverse of the Gaussian covariance matrix is the partial correlation. We will add these details to the appendix as you suggested.
>
> $\textbf{Response 3}$: Yes, the denominator will not be 0 because we modeled from the day where all areas had infectious subjects (this day was before the first day that we observed a case).
>
> $\textbf{Response 4}$: Please refer to Response 2 for the explanation.
>
> $\textbf{Response 5}$: Our model can accommodate reversal trends (e.g., waves of infections with multiple peaks) since the increasing or decreasing trend of infection rate $a(t)$ will be reflected in the waves of expected daily cases, and the first part of the loss function (Eq. 6) matches the observed and expected cases.
> Eq. 7 specifies the spatial dependence on each day $t$, where $\rho_t$ (the conditional dependence parameter) can be different for each day $t$ to allow time-varying spatial correlation. Similarly, $\rho_t$ can be different if the variance in infection rates changes along the course of transmission.
>
> $\textbf{Response 6}$: We used factor analysis because it does not require the factors to be uncorrelated with each other. This suits our medical application setting, and we found the fitted factors interpretable. If we require the factors to be uncorrelated, factor analysis will lead to the same model fitting as PCA. In fact, we found the fitted two factors have a low correlation, and thus  PCA is likely to yield similar results.
>
> $\textbf{Response 7}$: ${\hat {\bf {z}}_j}$ are the fitted factor scores, and they are part of the feature variables ${\widetilde {\bf {X}}_j}$.
>
> $\textbf{Response 8}$: For the first term of Eq. 6, we used $\{\sqrt{R_i(t)}-\sqrt{Y_i(t)}\}^2$ instead of $\{{R_i(t)}-{Y_i(t)}\}^2$ to stabilize the estimation. Square root transformation is a variance stabilization transformation for count data. When the observed case number is large, taking the square directly would enlarge the value too much, so we took the square root first and then calculated the squared error. We will explain it in the paper as you suggested.
>
>
> $\textbf{Experiments}$
>
> $\textbf{Response 1}$: Yes, we will plot the spatial map of the minority and mobility and provide them in the paper.
>
> $\textbf{Response 2}$: It is $\beta_t$, the coefficient for the effect of minority percentage on infection rate. Hence the result suggests that higher minority density is associated with a higher infection rate with statistical significance.
>
> $\textbf{Response 3}$: Here we describe the procedure to construct the confidence intervals for the spatio-temporal model. We subtracted the estimated from the observed number of daily new cases to get the residuals. We permuted residuals of each area across time, more specifically, exchanging residuals within days 1-14, 15-98, 99-160 as the variance of the residuals had a similar scale within each period. We did not permute across areas as it might disturb spatial correlation.
> We permuted 100 times and fit a separate model to each set of permuted data. Based on the permutation results, we estimated the variance of the parameters and constructed confidence intervals as estimate +/- 1.96*SE. We will add these to the appendix as you suggested.
>
> $\textbf{Response 4}$: Yes, the connection between the spatio-temporal model and the individual risk assessment model is through $w_{2j}$ and $M(t-7)$. We have run the model without $w_{2j}$ and $M(t-7)$. The model fitting and calibration were worse (AUC on the validation sets decreased from $0.776$ to $0.768$), which is as expected since the inferred number of infectious subejcts $M(t-7)$ was shown to be strongly associated with hospitalization in our full model with time-varying effects (shown in Figure 3a). Additionally, the fitted coefficients were different for other risk factors. For example, having diabetes was found to be associated with higher risk of hospitalization but without statistical significance (fitted coefficient: $0.06$, p-value: $0.78$). These results all demonstrate the utility of incorporating the inferred community factors in modeling the individual risk by correcting the selection bias and being a predictor.
>
>
> $\textbf{Response to Minor Comments:}$ We will correct the typos as you pointed out. Here is the link for the IRB under FDA regulations, https://www.fda.gov/regulatory-information/search-fda-guidance-documents/institutional-review-boards-frequently-asked-questions#IRBOrg. We will post our organization's IRB website if the paper is accepted.
>
>
> $\textbf{Response to Broader Comments:}$ As explained in Methods Response 2, jointly training data from all areas and integrating them with a spatial model will help to borrow strength from neighborhood areas, especially for the areas with low number of infections (their estimation will have high variance). One limitation of mis-specifying a joint model is that if the spatial dependence is low for certain areas, joint training may affect the individual area estimation. In our experiments on New York City (NYC) data, we detected a significant spatial correlation, and the proposed spatial model is beneficial for most of the zip areas. Future studies may extend the current model to allow a more flexible spatial dependence structure, e.g., community-specific dependence parameters. We will add the discussion to the paper.
> As explained by Methods Response 5, our model can naturally accommodate waves of infections. For forecast, our method can project daily case numbers under specified or projected case rates (e.g., assuming current trends of covariates and case rates will continue in the short term) and historical underlying infectious subjects estimated from the model.

---

> > ### Comment · Reviewer_e3RN · 2021-08-10
> > **Thanks for your response!**
> >
> > Thanks for your response! It has been really helpful in further understanding the proposed idea.
> >
> > I think the request of other reviewers in covering the prior works (and downplaying) on Spatio-temporal modelling is a reasonable one. However, I still think that this work pushes the boundary of merging the epidemiological analysis with socio-economic factors. This has significant relevance in policy decisions, and this work could be very useful to address or understand social inequities.

---

> > > ### Author Response · Authors · 2021-09-01
> > > **Thanks and author response**
> > >
> > > We thank the reviewer very much for all the questions and comments! They have led to a better version of our paper. As suggested, we will restructure the Introduction section to include more references and state our novelty and contribution more precisely. As you commented, our work aims to inform policymaking by studying social inequities in both COVID transmission and individual risk assessment, with careful consideration of spatial dependences and selection bias adjustments. We believe our approach is one step further to the current literature, and we sincerely appreciate your acknowledgment of our contribution.

---

### Official Review · Reviewer_L2tY · 2021-07-16

**Rating:** 5
**Confidence:** 4

**Summary:**

This study aims to estimate the daily infection rate and individual risk of hospitalization through offering a modeling of COVID-19 pandemic. They find correlations between the individual characteristics (e.g., age and race) with the community neighborhood. They have use real-world data of NYC cases during the 2020 pandemic.

**Limitations And Societal Impact:**

See above.

**Main Review:**

- My main concern with this paper is its relevance to the NeurIPS community. The paper is more suited for submission to data analytics and knowledge discovery venues.

- The topic of this paper is among one of the most studied during the past year. Modeling the Covid-19 and estimating the new cases are well-studied and there are plethora of models to benchmark the results of this study against. However, authors do not perform a thorough research on the related studies and offer no comparison with any benchmarks. It is hard to judge the merit of the proposed model in the absence of any baseline.

- The study offers marginal novelty over the existing methods, as evident from the amount of methods borrowed in the paper from the previous studies. More specifically, the impact of communities on a viral spread (and especially on COVID-19) is well-studied. So is the relationship between the individual features and the risk of severe outcomes upon diagnosis.

- Although the study claims to perform casual inference, the results only find correlation between the selected features. Moreover, the authors do not discuss any of the discovered correlations in detail (for example, what might be underlying cause of increased hospitalization among minority groups). The study does not offer much more than a simple analytic study.

- In figure C.3 in appendix where the difference between observed and predicted daily cases are given, for some zip codes the prediction is much weaker than the observed value (e.g., 11375, 11368, 11229). These cases are the interesting ones that show an underlying factor(s) change the pattern of infection unexpectedly. Deeper analysis to find the difference between these zipcodes and the rest help to understand the possible shortcomings of the proposed model (and addressing them).


Minor comments:
    - Line 24: "health system" --> "healthcare system"

**Time Spent Reviewing:**

3

---

> ### Author Response · Authors · 2021-08-10
> **Author Response**
>
> We thank you very much for all the constructive comments. Our responses are detailed in the following.
>
> $\textbf{Comment 1}$ is the main concern from the reviewer: “My main concern with this paper is its relevance to the NeurIPS community.”
>
> $\textbf{Response}$: COVID-19 is an ongoing pandemic still posing serious public health challenges. Machine learning and artificial intelligence have been playing a significant role in applications to respond to COVID-19 pandemic, from disease modelling, forecast, diagnostics, to risk prediction at individual, community, state and nation levels. In fact, on March 17, 2020 the White House has called for the development of new artificial intelligence techniques that can help researchers answer key questions about COVID-19 (https://healthitanalytics.com/news/white-house-urges-ai-experts-to-develop-tools-for-covid-19-dataset). To promote this goal, the White House and leading research groups have prepared the COVID-19 Open Research Dataset through the Kaggle platform.
>
> The NeurIPS community is well known to lead the field of machine learning and artificial intelligence, and an ideal venue to advocate for the use of novel methods to applications such as COVID-19 pandemic. The NeurIPS 2021 Call for Papers Guidance listed "Applications" as one of the topic areas, as well as stating "we welcome interdisciplinary submissions that do not fit neatly into existing categories, as well as work that addresses the social impact of machine learning”. As a matter of fact, NeurIPS 2020 accepted many papers that shared the same motivation as our work to analyze COVID-19 studies. For example, one accepted paper at NeurIPS 2020 was cited in our paper: "Interpretable sequence learning for COVID-19 forecasting".
>
> Therefore, we think that our work, which integrated machine learning and statistical methods and conducted multi-level analyses to study risk assessment for COVID patients, is highly relevant to this community.
>
>
> $\textbf{Comment 2}$ is about comparison with baseline methods.
>
> $\textbf{Response}$: For the individual risk assessment model, we compared with the model where no selection biases were corrected. Superiority was demonstrated by both model calibration and results that are consistent with the literature. For estimating COVID-19 new cases, we have reviewed several typical types of methods in the Introduction Section and will add more references as suggested. However, as discussed in the paper, all existing models only focus on forecasting area-specific COVID propagation, and no covariate is incorporated to study the factors that affect COVID transmission. Our method, on the other hand, aims to study the spatial correlation and the differential transmission patterns across communities as well as the influential factors which can guide the design of public health interventions.
>
> Furthermore, our proposed spatio-temporal model was built on a temporal model proposed in Wang et al. 2020 (reference provided in the paper) by integrating it with a spatial model inspired by the spatial disease mapping literature. The temporal model has been evaluated and is one of the models included in the CDC ensemble modeling group, which provides data-driven inputs to the CDC every week (https://www.cdc.gov/coronavirus/2019-ncov/science/forecasting/forecasts-cases.html). The focus in the current paper is to investigate whether one can accurately estimate the parameters of interest to inform public health policy makings, i.e., the time-varying effects of the community-level factors (e.g., race and mobility) on the disease transmission. In Supplementary Material Section C, we have shown that we can recover the true parameters of interest, and the MSEs were low.
>
>
> $\textbf{Comment 3}$ is about the novelty of this work.
>
> $\textbf{Response}$: Our proposed methods are novel in the following aspects:  We incorporated both individual covariates and spatial dependence to the individual-area temporal model developed by Wang et al. (2020), we adopted time-varying effect of community viral load into risk models, and we addressed selection bias (i.e., non-random sample of tested positive patients) embedded in the COVID electronic health records (EHR) data. The latter, although common to almost all COVID studies using EHRs, has never been addressed in the previous studies. Thus, our method, although built on some existing models, is not a direct application of these methods. The novel way in the methods allows us to better answer questions that were not well studied in the literature, e.g., the racial effect on COVID transmission and selection biases in the EHRs.
>
> Our findings are also new. For example, we found that a higher minority percentage was significantly associated with higher infection rates, and such effect varied along the course of the COVID outbreak. We also detected significant interactions between an individual’s race and community minority percentage (and multi-unit living environment) on hospitalization. These finding can inform policy makers on what communities, which populations, and when to target intervention to reduce COVID transmission and hospitalization. We provide more detailed discussions on this point in the response to Comment 4.
>
>
> $\textbf{Comment 4}$ is about causal inference.
>
> $\textbf{Response}$: For observational data, under no unmeasured confounding and sufficient adjustment for selection bias assumptions, we can draw causal inference based on the individual risk assessment model. We emphasize causal inference because we adjusted for multiple selection biases and included broader confounders (not limited to individual characteristics) as covariates in the individual model. Without addressing these biases, the findings can be misleading as we have shown in the unweighted analyses. The increased hospitalization among minority groups can be due to other social determinants of health (not collected in the data), including socioeconomic status, access to health care, and exposure to the virus related to occupation, e.g., frontline, essential, and critical infrastructure workers (we will add references and discussion to the paper).
>
> Furthermore, our causal model results have actionable implications on designing precision public health policies. For example, the detected significant interaction between an individual’s race and community minority percentage on hospitalization can guide policy makers on how to target intervention to reduce hospitalization burdens, for example, target Hispanic and black communities living in areas with dense minority populations. Similarly, the detected significant interaction between an individual’s race and his/her multi-unit living environment suggests targeting the intervention for multi-unit living buildings specifically for Hispanics. While individual-level risk factors are difficult to intervene on, these neighborhood-level factors provide pathways for designing tailored interventions.
>
>
> $\textbf{Comment 5}$ is about Figure C.3.
>
> $\textbf{Response}$: In Figure C.3, for most zip-code areas the estimated cases matched well with the observed, but for a few zip-code areas, the number of cases was under-estimated, especially around the peak period due to multiple reasons. First, the data variability was high in some zip areas where there were abnormal/extreme spikes. Those spikes were primarily due to data backlog and sudden “data dumps” instead of true case rises. Since our model encourages a smoother fit by incorporating penalty terms for consecutive days, those abnormal spikes cannot be (and perhaps should not be) captured. Second, since we pool information from neighborhood areas to infer infection rates, when the local spatial dependence is not very strong, pooling information may affect the area-specific case estimation. However, for the majority of zip areas, the proposed spatial model is beneficial. Future studies may extend the current model to allow more flexible spatial dependence structure, e.g., community-specific dependence parameters.

---

> > ### Comment · Reviewer_L2tY · 2021-08-28
> > **Re-evaluation**
> >
> > Firs, I appreciate the authors' thorough response to all reviewers, it answered majority of my concerns.
> >
> > (1) Now I understand the main contributions of the paper better. As reviewer KY3s mentioned, I had the impression that the main novelty of the paper lies in the spatio-temporal model used for the community risk assessment (which indeed is not novel). Based on the authors' response, the novelty of the work lies in the the mixture of (already known) community level and individual level factors that contribute to risk of hospitalization and death among COVID patients.
> >
> > (2) That being said, I don't agree with some of the claims made by authors. For example, with respect to *"Our findings are also new. For example, we found that a higher minority percentage was significantly associated with higher infection rates"*, numerous studies have already shown such phenomenon using different datasets. A very small fraction of them are:
> >
> > https://www.ncbi.nlm.nih.gov/pmc/articles/PMC8248751/   [dataset used: mixed]
> > https://www.healthaffairs.org/doi/10.1377/hlthaff.2021.00098  [dataset used: California]
> > https://www.thelancet.com/article/S0140-6736(21)00634-6/fulltext  [dataset used: England]
> > https://academic.oup.com/cid/article/72/5/e88/5998295?login=true  [dataset used: Michigan]
> >
> > Indeed, the findings highlighted by the authors to support the novelty of their findings are exactly those found and discussed in more depth in https://www.healthaffairs.org/doi/10.1377/hlthaff.2021.00098 (e.g., racial disparity, high-risk-household as defined by the number of essential workers living there and the ratio of rooms to inhabitants). Obtaining the same conclusion as the previous studies enforces the validity of the model, but does not contribute much to proving its superiority in designing public health policies.
> >
> > In general, I feel the emphasis on the **practical** implications of their proposed models is not backed up by enough evidence, either in the form of comparison with current models in effect or in the form of a comprehensive case study and interpretation of results (for example, the possible underlying factor(s) in racial disparity).
> >
> > (3)  Comment (5) and similar possible drawbacks of the model (and their potential solutions) should be mentioned as limitations (section 4).
> >
> > Considering (1), I am willing to upgrade my evaluation. However, I am still not convinced about the practical merits of the proposed model (as mentioned in (2)). Furthermore, I agree with reviewer KY3s regarding the changes that need to be made in keeping the claims on novelty fair and I also think the authors should dive deeper into analysis of their results as the most exciting part of the paper. As such, I don't see the paper in its current format to be a great fit for NeurIPS and would change my evaluation to "marginally below the acceptance threshold".

---

> > > ### Author Response · Authors · 2021-09-01
> > > **Thanks and author response**
> > >
> > > We thank the reviewer very much for the re-evaluation of our paper and willingness to upgrade the evaluation. We appreciate these inputs which have led to a broader discussion of our work.
> > >
> > > Regarding comment (1), we appreciate that reviewer now recognizes our novelty on the methodology side which involves the spatio-temporal modeling with a mixture of community-level and individual-level risk factors, and provide individual risk assessments. Besides, another novelty as stated in the paper is to adjust for selection biases intrinsic to EHR data of COVID patients using the latent community viral loads as a time-varying risk factor. As the reviewer suggested, we will restructure the Introduction section to state each point of our novelty more precisely.
> > >
> > > Regarding comment (2), our work leads to multiple interesting findings, and “a higher minority percentage was significantly associated with higher infection rates” is only one of them. More interestingly, our method detected both the main effects and interactions between community- and individual-level risk factors and revealed the time-varying effects on COVID hospitalization. In contrast, the epidemiologic findings in the literature referenced  by the reviewer and in all other related literature we have searched did not have such results and did not adjust for selection bias.
> > >
> > > We provided practical implications of our findings in the previous response. For example, a significant interaction can facilitate precision public health decision-making at both community- and individual-level, i.e., to inform when, which population and in what communities should we target the intervention to better reduce the hospitalization burden. The finding of the racial disparity may prompt policymakers to further look for possible reasons and solutions for this issue.
> > >
> > > For comment (3), we will describe more clearly in the paper the assumptions of our spatio-temporal model and state the potential limitations of the joint modeling approach when the assumptions are not met. We will also discuss the solutions, for example, to extend the current model to allow for a more flexible dependence structure. The detailed discussions of the advantages of the joint modeling approach and the solutions were provided in our previous responses ( “Methods Response 2” and “Response to Broader Comments” to Reviewer e3RN).

---

### Official Review · Reviewer_KY3s · 2021-07-26

**Rating:** 6
**Confidence:** 5

**Summary:**

Authors propose a spatio-temporal model for modelling evolution of SARS-CoV2 and then they use that model with the a semi-parametric model to analyse risks of COVID-19 at individual levels.

**Ethical Concerns:**

none they have permision from IRB

**Limitations And Societal Impact:**

Yes they have

**Main Review:**

Authors propose a spatio-temporal model for modelling infections, hospitalisations and deaths related to COVID19. They then further extend their analysis by using a semi-parametric model to asses individual level risks by using EHR data from two hospitals in NYC. The  paper is written in a very elegant and simple way that makes it easier for reader to parse through the ideas and get to the main goals. The experiments and methodology is sound.

I think, authors are overplaying their contributions in terms of coming up with a spatio-temporal models. Temporal models are back bone of infectious diseases, they are not just limited to citation 26 in the paper. The history of models using delay distributions could go way back to use of renewqal equations in Willy Feller. “On the Integral Equation of Renewal Theory”. In:The An-nals of Mathematical Statistics(1941).issn: 0003-4851.doi:10.1214/aoms/1177731708 or Richard Bellman and Theodore Harris. “On Age-Dependent Binary Branch-ing Processes”. In:The Annals of Mathematics(1952).issn: 0003486X.doi:10.2307/1969779. Practical use of these equations could be seen in Fraser, C. Estimating Individual and  Household Reproduction  Numbers in an Emerging
Epidemic. PLOS ONE 2, e758 (2007)., E. Goldstein et al.: Proc. Natl. Acad. Sci. U.S.A. 106, 21825 (2009) and A. Cori et al.: Am. J. Epidemiol. 178, 1505 (2013). In fact one of the very first model for modelling covd19 in Europe, S. Flaxman et al.: Nature 584, 257 (2020),  is also based on approach the authors claim to be novel. I would urge them to downplay this claim and even the claim that infectious disease models generally are not temporal, at best it is actually exactly the other way around. Similarly various spatial models have also been around there for modeling SARS-COV2 for long like J. S. Jia et al.: Nature 582, 389 (2020), Giuliani, D., Dickson, M.M., Espa, G. et al. Modelling and predicting the spatio-temporal spread of COVID-19 in Italy. BMC Infect Dis 20, 700 (2020). https://doi.org/10.1186/s12879-020-05415-7, F. Bartolucci et al.: Spatial Stat. 100504 (2021), https://localcovid.info/data-methods.html and many others. So again I willask authors to downplay their claims.

Similarly like modelling of spread the model to account for individual level risk  is standard and something used in the public health for a long-long time. SO again these contributions are not novel, hence request is to downplay this part.

However, the results and analysis is quite detailed and looks for things of interest. I will urge authors to talk about them more and push that as their main contribution. Lastly, I will like authors to release their code at least. I can see why data they can't but see no reason behind not releasing the code.

**Time Spent Reviewing:**

5

---

> ### Author Response · Authors · 2021-08-10
> **Author Response**
>
> We thank you very much for all the positive comments and constructive suggestions. Our responses are detailed in the following.
>
> Following your suggestion, we will focus more on our novel contributions in the context of the spatio-temporal modeling and provide a more comprehensive review of the literature including those mentioned by you. For example, the existing spatial and temporal models for COVID-19 only focused on forecast, and they did not incorporate contextual covariates to model the disease transmission. We will also highlight that our proposed spatio-temporal model allows examining the effects of area-specific covariates reported by the CDC (e.g., mobility and percentage of minority) on the disease infection rates. In addition, some existing methods model individual-area temporal trends and spatial patterns separately. By jointly modeling the temporal and spatial patterns across all areas, our approach allows explaining what accounts for the spatial variability and the spread of infectious disease, and would be more efficient than modeling two trends and for different areas separately. Our results showed a significant racial disparity and significant spatial correlation.
>
> We have reviewed a few typical temporal models used for forecasting COVID cases in the Introduction Section, and we will add the references suggested by you for a more complete review.
>
> For the individual risk assessment models, although logistic regression model is standard in public health research, we chose this model because it usually fits data reasonably well, is widely accepted in practice, and allows easy interpretations of the risk factors that can be understood by policy makers. One of the new contributions here is to introduce time-varying coefficients of the community-level factors to capture their changing effects depending on the stage of the pandemic and the evolving nature of COVID-19. More importantly, using the inferred community-level factors (i.e., the estimated prevalence of true infections), we can mitigate the selection bias intrinsic to the study sample of patients from Electronic Health Records (EHRs) data who tested positive. Such selection bias is common to almost all studies based on non-random sample of tested positive patients but was not addressed in other public health research of COVID. As demonstrated in the paper, our experiment on the NYC hospital data showed that correcting the selection biases led to better model calibration on the validation sets and more meaningful results. In particular, after correcting for the selection bias, diabetes was found to be significantly associated with a higher risk of hospitalization, which is consistent with the literature and CDC guidance. In contrast, without correcting for the selection bias, diabetes was found to lower the risk of hospitalization, which is unexpected and inconsistent.
>
> As you suggested, we will also focus on the results and their implications on designing precision public health policies. In our experiment on NYC data, we found that a higher minority percentage is significantly associated with higher infection rates in the spatio-temporal model, and such effects are time-varying along the course of the COVID outbreak. This result can potentially inform policy makers on which communities and when to target interventions to reduce COVID transmission. For the individual risk assessment model, we detected a significant interaction between an individual’s race and the community minority level on hospitalization, which can guide the design of targeted public health intervention to reduce hospitalization burden, for example, by targeting Hispanic and black communities living in areas with dense minority populations. Similarly, we detected significant interaction between race and multi-unit living: target the intervention to multi-unit living buildings specifically for Hispanic population. While individual-level risk factors are difficult to intervene on, these neighborhood-level factors provide pathways for targeted interventions to relieve severe COVID outcomes in individuals. We will add all these discussions to the paper. Additionally, we will add figures to better visualize the variability in the community-level factors, such as minority density and mobility across NYC zip-code areas.
>
> The program code was already submitted in the last round as separate files in supplementary materials, including codes to create the simulated data and codes to run the experiments.

---

> > ### Comment · Reviewer_KY3s · 2021-08-31
> > **Thanks scores kept same**
> >
> > I will like to thanks the authors for their response to all reviewers in detail. However, I think I still do not see what are the major contributions of this work apart from already known and published, unless authors dive dip into their findings. A small passing by comment is that both mobility and social covariates are something that has been part of epidemic models (even COVID ones) for long now and authors still seem to claim its use as a contribution which I disagree with.
> >
> > The response doesn't seem to shed light on my comment about diving deep into the results. Also, I do agree with reviewer L2tY that they need to do more experiments to substantiate the performance or the results.

---

> > > ### Author Response · Authors · 2021-09-01
> > > **Thanks and author response**
> > >
> > > We thank the reviewer very much for all the inputs, which lead to a broader discussion of our paper. Apart from the Spatio-temporal model, another contribution of our work is to investigate the individual risk for COVID hospitalization, adjusting for the selection biases intrinsic to EHR data of COVID patients using latent community viral loads as a time-varying risk factor. Moreover, our method detected both the main effects and interactions between community- and individual-level risk factors and revealed the time-varying effects on COVID hospitalization. Such selection bias though common was not addressed in current risk models, and the time-varying and interaction effects were not well studied in the current literature.
> > >
> > > Our findings have practical implications and here we provide more discussions and interpretations. For example, a significant interaction and time-varying effect can facilitate precision public health decision-making at both community- and individual-level, i.e., to inform when, which population, and in what communities should we target the intervention to better reduce the hospitalization burden. For example, it can be of higher utility to target the intervention to Hispanic and black communities living in areas with dense minority populations and target the multi-unit living buildings specifically for the Hispanic population. The increased hospitalization among minority groups can be due to other social determinants of health (not collected in the data), including socioeconomic status, access to health care, and exposure to the virus related to occupation, e.g., frontline, essential, and critical infrastructure workers. The interesting interaction effects we detected may reflect the interplay among racial, socio-economic and behavioral factors. For example, it’s likely that Hispanic and black people living in the non-minority communities are those wealthier individuals who pay more attention to personal hygiene and COVID risk prevention.
> > >
> > > In the experiments, for the individual risk assessment model, we compared it with the model where no selection biases were corrected. Superiority was demonstrated by both model calibration and results that are consistent with the literature. For the Spatio-temporal model, our goal was to test whether one can accurately estimate the parameters of interest to inform public health policy-making, i.e., the time-varying effects of the community-level factors (e.g., race and mobility) on the disease transmission. In Supplementary Material Section C, we have shown that we can recover the true parameters of interest, and the MSEs were low.

---

### Comment · Area_Chair_6JH7 · 2021-08-19
**Discussion**

Thank you to the reviewers and authors for initiating an interesting discussion around this manuscript. This manuscript presents a very timely discussion around understanding how ML can be used to understand covid risk as a pathway towards optimising the public health decision-making process. In light of the authors' response, as well as the comments provided by other reviewers, I would be interested to hear whether reviewers KY3s and L2tY think that their comments have been adequately addressed and whether the authors' rebuttal warrants a change in the original score assigned.

---

### Decision · Program_Chairs · 2021-09-27

**Decision:**

Accept (Poster)

**Comment:**

This paper presents novel ML approaches to examining the socioeconomic disparity of infection risks and interaction among the risk factors, to assist public health decision-making and facilitate better clinical management of COVID patients. This study is highly relevant to the NeurIPS community. As one reviewer states, this work pushes the boundary of merging the epidemiological analysis with socio-economic factors and is quite ground breaking in this respect, with the potential for huge societal impact. The paper is methodologically solid and the authors have presented a strong rebuttal and discussion with the reviewers